# Justice in Dementia Care Resource Allocation: How Should We Plan for Dementia Services?

**DOI:** 10.3390/ijerph16101754

**Published:** 2019-05-17

**Authors:** Chia-Feng Yen, Shyang-Woei Lin

**Affiliations:** 1Department of Public Health, Buddhist Tzu-Chi University, Hualien 97004, Taiwan; mapleyeng@gmail.com; 2Department of Natural Resources and Environmental Studies, National Dong Hwa University, Hualien 97401, Taiwan

**Keywords:** resource allocation, dementia, disability, geographic information system, justice, profit willing distance, tolerance limited distance

## Abstract

Dementia care resources in Taiwan have not been allocated taking into account patients’ needs and the distance between service users and providers. The objective of this study was to use two newly developed indicators; profit willing distance (PWD) and tolerance limited distance (TLD), to profile the service availability and accessibility of the 22 administrative areas in Taiwan and facilitate justice-based resource allocation by the central government. The study employed secondary data analysis by using a geographic information system (GIS) and geocoding to identify distances between service users and providers. The study samples were drawn from the databank of the National Disability Eligibility Determination System and grouped by the acuteness of registrants’ needs. Both the PWD and TLD were found in 15 of the administrative areas, and neither was found in three areas (Penghu, Kinmen, and Lienchiang County). Either the PWD or TLD (but not both) were found in four areas (only have PWD: Hsinchu and Chiayi City; only have TLD: Yunlin and Taitung County). How the priorities should be set for dementia service allocation based on these findings was also addressed. We conclude that the indicators of PWD and TLD can add value to the policy decision-making process, help set priorities, and facilitate efficient and fair resource allocation by defining specifics of the resources needed.

## 1. Introduction

According to the World Alzheimer Report 2018 and cross-national surveys, there were over 10 million new cases of dementia in 2017, with an average of one dementia case every three seconds. It was estimated that the number of patients with dementia would reach 131.5 million by 2050 [1,2]. It was estimated that the cost spent on dementia care in 2015 would have been USA $818 billion, and by 2018 it would exceed USA $1 trillion [1]. In Taiwan, the prevalence of dementia in 2013 was 8% among the population above 65 years old. In 2017, the total population of dementia was estimated to be over 270,000 which was larger than the total population of Changhua City (population: 235,000), and the population of dementia who would need care could reach more than 850,000 in 40 years [3]. However, given the limited resources of long-term care, the central government has the responsibility of ensuring that national resources for dementia care are allocated based on distributive justice in order to meet the needs of patients with dementia and their families [4,5,6].

The most commonly used indicators by countries for allocating public resources (such as medical or social services) include the population of demand, the ratio of medical staff to the population, and the number of medical centers or beds [7,8,9]. Lee and Lu developed the concentration index (CI) and the index of horizontal inequity (HI) to evaluate health inequity and the use of medical care by children and to examine the relationships between income and the use profile of care. They considered family income and children’s health status, but not burden of distance [10,11]. In recent years, many studies have utilized the spatial analysis of the geographic information system (GIS) to investigate patients’ care seeking behaviors [12,13,14,15,16], medical resource differentials between urban and rural communities [17,18,19], and indicators of medical accessibility (for example, the shorter the distance, the higher density of service providers) [13,20,21]. However, such an approach failed to capture reality on the ground and effectively solve the problem of inequality by studying the following factors separately, such as the number of users in need of services provided, distance between users and providers, user’s health status, and density of services and providers. This approach overlooked the interactive effects between factors, for example, the interactive effects between users and the distance to providers, as well as the effects between the number of services and user’s health status.

Hence, the purpose of this study was to use the modified accessibility indices of PWD (profit willing distance) and TLD (tolerance limited distance) to (1) describe and compare the accessibility of dementia services across administrative areas in Taiwan; and (2) provide evidence-based input to the central government to determine the priority areas for establishing service resources based on distributive justice [19]. PWD and TLD were developed and published in 2015 and used to distinguish national resource allocation more efficiently than other indexes. In their study, the definition of PWD was the capacity and willingness of providers to supply services at different distances (the definition of PWD here is different from the idiomatic initialism of people with dementia) and TLD was defined as the decrease in the number of providers that indicated the burden of supply over the distance, which were used to discuss home nursing care resource disparities in rural and urban areas.

## 2. Materials and Methods

This study employed secondary data analysis and drew data from the National Disability Eligibility Determination System (DEDS) in Taiwan, a nationwide registry of the population with disabilities. The system contains the following information: basic demographic data, residence status (in institutions or in communities), impairment profile (e.g., the body function and body structure based on the International Classification of Health, Functioning, and Disability, main ICD-9-CM codes of disability, and functioning evaluation data). The data were collected by 239 hospitals that were authorized to conduct disability evaluation in Taiwan. The evaluations were carried out by physicians and other professionals such as occupational therapists (OTs), physical therapists (PTs), speech therapists (STs), social workers, psychologists, and nurses.

The present study was approved by the Research Ethics Committee of the Hualien Tzu Chi Hospital, Buddhist Tzu Chi Medical Foundation (IRB102-178). The functioning evaluation of the adults with disabilities was conducted using the Chinese version (in traditional Chinese) of the 36-item version of the World Health Organization Disability Assessment Schedule 2.0 (WHODAS 2.0-36 item) [22,23].

### 2.1. Participants

Users of the community services for dementia participated in the study and comprised adults with disability 18 years or older who were officially registered in the DEDS in Taiwan from July 2012 to October 2013. The total population with disabilities was 157,478 during this period and 11,967 were diagnosed with dementia (ICD-9-CM codes: 290 to 331). With residents in institutions (*n* = 2263) excluded, 9704 patients with dementia living in the community were included in this study (Figure 1).

The dementia related-care providers were announced on the Web page of Ministry of Health and Welfare in 2016. These providers were legally authorized to provide such care. A total of 2116 providers of dementia care in Taiwan were identified, and 330 were excluded because they shared the same geographical coordinates. Finally, there were 1786 providers included in our study [24]. The dementia-related care rendered by these providers included home services, home respite care, home nursing care, home rehabilitation, daycare services, family care services, Dementia Elderly Group Homes, Veteran Houses and so on.

### 2.2. Materials

#### Definitions of General and High Level of Need for Dementia Care Services Based on the WHODAS 2.0-36-item

The WHODAS 2.0-36-item was developed based on the International Classification of Functioning, Disability and Health (ICF) of the WHO in 2010 to measure patients’ activities and participation in daily living in each of the following 6 domains within the previous 30 days: (1) cognition (six items), by assessing communication and thinking activities such as concentrating, remembering, problem solving, learning and communicating; (2) mobility (five items), by assessing activities such as standing, moving around inside the home, getting out of the home and walking a long distance; (3) self-care (four items), assessing activities such as hygiene, dressing, eating and staying alone; (4) getting along (five items), by assessing interactions with other people and any difficulty experienced due to health conditions; (5) life activities (eight items—pertaining to the household, school, or work), by assessing any difficulty experienced with day-to-day activities (activities that people perform on most days) which are associated with domestic responsibilities, leisure, work and school; and (6) participation (eight items), by assessing the social dimensions of the environment where the respondent resides such as community activities, barriers and hindrances, as well as problems encountered such as maintaining personal dignity. The possible responses to each item are: no difficulty, mild difficulty, moderate difficulty, severe difficulty and extreme difficulty [25]. The total score ranges from 0 to 100 and the higher the score, the disabled level more severe. The participants answered 32 items altogether, that is, the total of 36 items minus those related to employment and studying. The Chinese WHODAS 2.0-36-item was developed and published between 2013 and 2014 in Taiwan and has shown good validity and reliability [22].

To determine the acuteness in demand for dementia care services, we used the functioning status of the patient. Based on the methodology employed by Huang et al. (2015) who used the domain score of the WHODAS 2.0 to predict the need for institutionalization of individuals with dementia [26], we included cases in the current study whose scores were higher than the following cutoffs: Domain 1 score > 77.5, Domain 2 score > 78.5, Domain 3 score > 55, and summary score > 66.5 (Appendix A). The group with high levels of need for dementia services comprised 3111 cases and the group with general demand, 6593 cases.

### 2.3. Data Analysis

Data were analyzed using the Statistical Package for the Social Sciences (version 20.0, SPSS, Chicago, IL, USA), join point analysis and a geographic information system (GIS, ArcGIS 10.3, Esri, Redlands, CA, USA).

#### 2.3.1. Spatial Analysis

The GIS geocoding was first applied to convert the addresses of the provider and the user of community-based services for dementia into (x, y) coordinates. Each point is derived from a specific spatial process that involves a minimum number of administrative centers based on the zip code for ethical reasons.

The data were then plotted on a digital map and the nearest distance between the locations of the user and the provider was identified using the spatial join of the GIS. The supply of all resources (providers) and cases (users) within the same administrative area were located. The spatial join analysis matches the join feature with the target feature based on their relative spatial locations. A match was made between a provider’s and a target case’s locations when the nearest distance between the two was found. The distances between all cases and providers were determined accordingly. A continuous function was created where all the distances were inputted as the value of variable x, and the cumulated fixed intervals as the value of variable y. We take Taipei City as an example in Figure 2. All the nearest distances between the locations of the user and the provider have formulated some relationships.

#### 2.3.2. PWD and TLD Measurements: Indices of Accessibility of Services

We used join point analysis to define the PWD and TLD indices for the supply of dementia care. The original PWD and TLD were developed based on the concept of continuity, using regression and differential equations to define the PWD and TLD. This method involved cumbersome steps and was actually less reproducible [19]. In this study, we used standardization and a more succinct method to define these points, instead.

The PWD was defined as the distance which providers would accommodate to willingly supply dementia services to users. Service users’ distances from providers were grouped based on the measurement unit of 50 m. Join point analysis was used to identify the first significant inflection point—defined as the “PWD”, that is, the distance associated with the maximum number of providers.

The TLD was defined as the distance between providers and service users which was within the range of users’ burden to use. In other words, providers and the services rendered beyond the TLD were simply too far away from the users. The distances between the significant inflection point and participants’ locations were computed.

Inflexion points were determined by join point analysis, and there could be more than one inflexion point; as Figure 2 shows, there were many inflexion points in this relationship. Inflexion points 1 and 2 indicated the first significant points which marked the nearest distances from the origin. The PWD was defined by the first significant point where the slope changed from positive to negative, and the TLD was defined by the first significant point where the slope shifted from negative to positive.

## 3. Results

### 3.1. Characteristics of Service Users and Providers

The mean age of the cases was 78.2 years old (SD = ±9.8) and 63% of them were female. Based on the Taiwan Disability Evaluation System, the severity of disability among the cases ranged as follows: 28.1% (mild), 37.3% (moderate), 7.6% (severe), and 27.1% (extremely serious). The group with general need for dementia care was significantly younger and suffered from milder forms of disability than the group with high level need for dementia services (Table 1).

Table 2 presents the WHODAS 2.0 scores of the service users. All the domain scores for the high level of need group were higher than those of the general demand group (*p* < 0.001). In the high level of need group, all domain scores were above 90 and the highest score was seen in Domain 5 of “household activities” (score: 99.37). This means that the service users needed support by others or assistive devices almost every day for activities and participation.

There were 1782 social service units with different geographical coordinates nationwide that were designed specifically to provide dementia care, and the overall mean of service densities (the ratio of people with dementia to providers) was 5.45. The service densities of seven cities and counties were higher than the overall means (Table 3). Table 3 also shows the resource allocation in accordance with the service density among the dementia population.

### 3.2. Service Availability/Accessibility and Burden of Distance

In the present study, we measured the PWD to examine where the availability and accessibility of dementia care were high in Taiwan. In Taipei, the PWD was 650 m, the highest in Taiwan, which means every provider in the city was willing to offer dementia services within 650 m. In other words, dementia services were readily available and accessible to users in need of such services. Aside from Taipei City, high PWDs were also found in Hualien County (600 m) as well as Tainan City and Kaohsiung City (350 m). There were seven cities and counties with the lowest PWD (150 m) (Table 4). There were five administrative areas without PWDs, which was indicative of no dementia services available or accessible to patients with dementia.

Nationwide, the lowest TLD was 1200 m and the highest was 6250 m. The highest TLD was found in Yunlin County. The TLD of each city and county represents the threshold of justice in resource allocation and should be the priority distance considered by the government when setting up dementia resources. There were five administrative areas without TLDs. No PWDs were found in three of them (Penghu, Kimen and Lienchiang Counties), either (Table 4).

There are a total of 22 administrative areas (counties and cities) in Taiwan. Both the PWD and TLD were found in 15 areas, and neither was found in three areas. Either PWD or TLD (but not both) was found in the following four administrative areas: Hsinchu City and Chiayi City, with PWD but not TLD; and Yunlin County and Taitung County, with TLD but not PWD (Table 5). In Figure 3, part of New Taipei City is enlarged to illustrate the different geographic categories also seen in the other 14 administrative areas with both the PWD and TLD. The geographic categories include areas with resource supply, profitable areas, tolerable areas, and areas beyond the tolerance zone. The remaining cities/counties charted in Figure 3 include the four areas with either PWD or TLD, but not both. The indices (PWD and TLD) for these five areas are also displayed in the figure.

## 4. Discussion

The results of the present study shed light on the priority areas (cities or counties) where national welfare services (dementia resources) should be set up. The PWD and TLD appear to add more value to support the policy-making process than some traditional indices which focus primarily on supply and demand by examining factors such as user–service ratio, population in need of services, and number of providers. The PWD and TLD were developed on the basis of not only supply and demand but also spatial autocorrelation. Whether or not existing resources are sufficient can be determined by measuring the PWD and TLD and comparing the TLDs of different administrative areas to guide the central government on where to set up related resources.

### 4.1. PWD & TLD vs. 2SFCA & E2SFCA

In 2005, Wang & Luo developed the 2SFCA (two-step floating catchment area) method to evaluate the accessibility of medical care services in different administrative areas. The method considered spatial and non-spatial factors such as the individual’s age, gender, race, socioeconomic status and language skills, as well as the characteristics of a population or area such as land use, university graduation rate, ratio of single-parent families, state of unemployment, and ratio of occupational categories. The indicators of health needs and service accessibility were then calculated by integrating these factors and assigning weights based on the relative eigenvalues of these factors [27]. This method was believed to yield more accurate measurements. In 2009, McGrail & Humphreys published a similar index which was developed based on the 2SFCA method to assess the accessibility of primary care in rural Australia and verify the feasibility of this method. They found that this weighted method was based on fuzzy logic and that all variables, with the only exception of service density, were non-spatial indicators. Collecting such information was time consuming and costly, although the researchers did acknowledge that the 2SFCA method yielded highly accurate results [28]. Later, Luo & Qi (2009) and Kilinc et al. (2016) developed the E2SFCA (enhanced two-step floating catchment area) method by adding spatial cluster analysis and trying to simplify the 2SFCA method. In the end, however, they were still unable to simplify the calculation process [29,30]. Compared with the PWD and TLD (Table 6), not only does the issue of fuzzy weight basis remain with the 2SFCA and E2SFCA methods, but it is also difficult to convince the public and decision makers when using these methods.

In 2015, Lin et al. (the same research team as the current study) developed new indices, PWD and TLD, to demonstrate, based on the burden of distance, the threshold of provider’s capacity for offering nursing home care. However, the authors also noted that the PWD and TLD were defined by running differential equations multiple times through cumbersome steps and cautioned about the reproducibility of this method when applying it to other studies [19]. In the present study, improvement was made to the method of detecting the curve break and to make it easier to apply the method to other studies. Our study found that the materials for indicator development were more readily accessible for the method of PWD and TLD, that the method was easier to use to support policy decision-making than the 2SFCA method, and that the method of PWD and TLD yielded more accurate results than traditional indicators.

### 4.2. Policy Priorities When Setting up Dementia Services

In terms of the priority when setting up dementia service resources (Table 4), the areas with only TLD but not PWD (Yunlin and Taitung County) should be considered first. For these areas, dementia service users’ needs remain unmet although users still try to find services as much as they can. Next, the consideration should be directed toward the areas with both the PWD and TLD, with priority given to those with higher TLDs. Lastly, areas with only PWD but not TLD and areas where neither was found should be considered.

In addition, users’ needs for dementia care should also be factored in. Given the limited resources to support an increasing population with dementia and the difficulty in meeting the needs of all patients with dementia through existing national welfare programs, meeting needs of the high level of need group should be the priority. As demonstrated by the functioning scores of the WHODAS 2.0-36 items in the present study, there were 3111 dementia patients in this group who experienced difficulty in performing daily life activities and participation. The government must meet their care needs based on the principles and mandate of social welfare that the same with other researchers mention in other studies [31,32].

In the current study, neither the PWD nor TLD was found in the administrative areas of Penghu, Kinmen and Lienchiang Counties for all participants as an aggregate whole. The same findings were mirrored when the general demand group and the high level of need group were examined separately. These findings were not a total surprise, given that all three counties are remote administrative districts in Taiwan (i.e., offshore island counties) where less supply and demand is expected. When the general and high level of need groups was studied separately, we observed that the fewer number of cases, the more administrative areas where both indicators couldn’t be found. We speculate that the sample sizes might be too small and that the regression analysis and differential equations based on the study of Lin et al. (2015) might be preferred to the join point analysis employed in the present study [19]. Furthermore, it should be careful that to apply the methods of this study, traveling time is more appropriate than the nearest distance to measure PWD and TLD. Especially so in urban areas, where the availability of transport may have an effect.

In the future, to implement justice-based allocation of national social services, it will be useful to capture the reality on the ground by measuring the indicators of PWD and TLD. The administrative areas as profiled by the PWD and TLD can then be compared to identify those where resource deficiencies reside.

## 5. Conclusions

The two indicators of PWD and TLD developed in this study consider not only the availability of care services (by providers) but also the burden of distance between the service user and the service provider. The concepts of PWD and TLD are very important particularly in some countries, their public medical services fee paid by the government don’t include the cost due to distance which must be paid by providers or users. The distance cost indeed influences the willingness of providers and the accessibility of national welfare services. Most importantly, the PWD and TLD can add value to the policy decision-making process, help set policy priorities, and facilitate efficient and fair resource allocation by defining the specifics of the resource needed.

## Figures and Tables

**Figure 1 ijerph-16-01754-f001:**
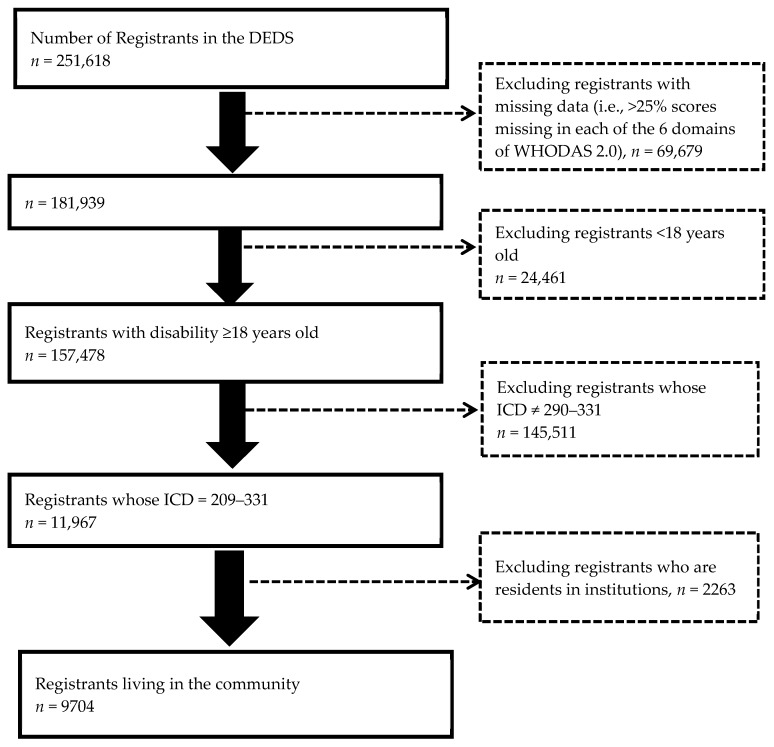
The sample selection process. DEDS = National Disability Eligibility Determination System (Taiwan).

**Figure 2 ijerph-16-01754-f002:**
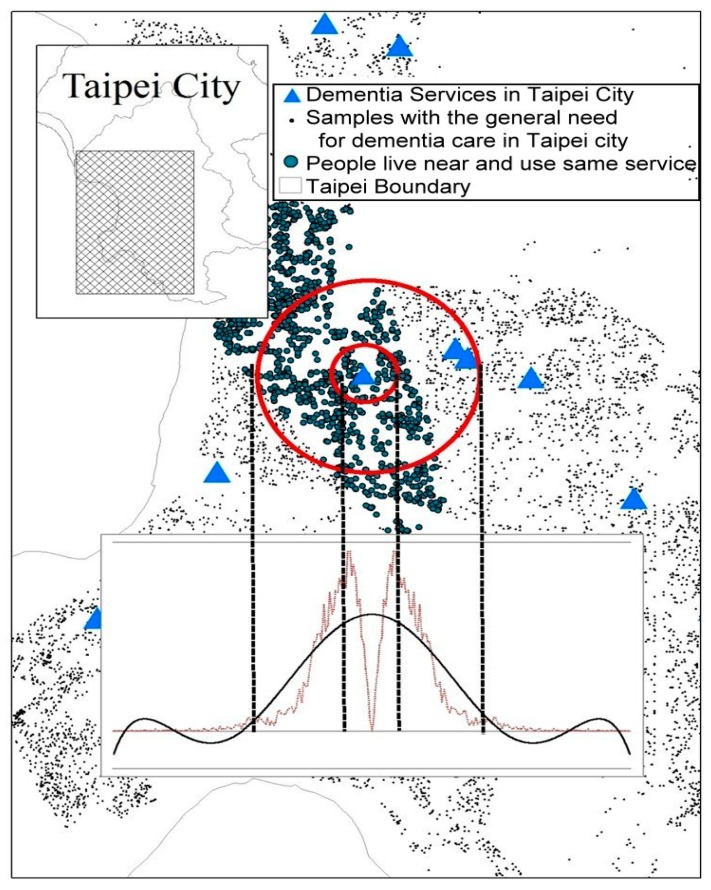
Relative spatial relationship between service users and providers: Taking Taipei City as an example.

**Figure 3 ijerph-16-01754-f003:**
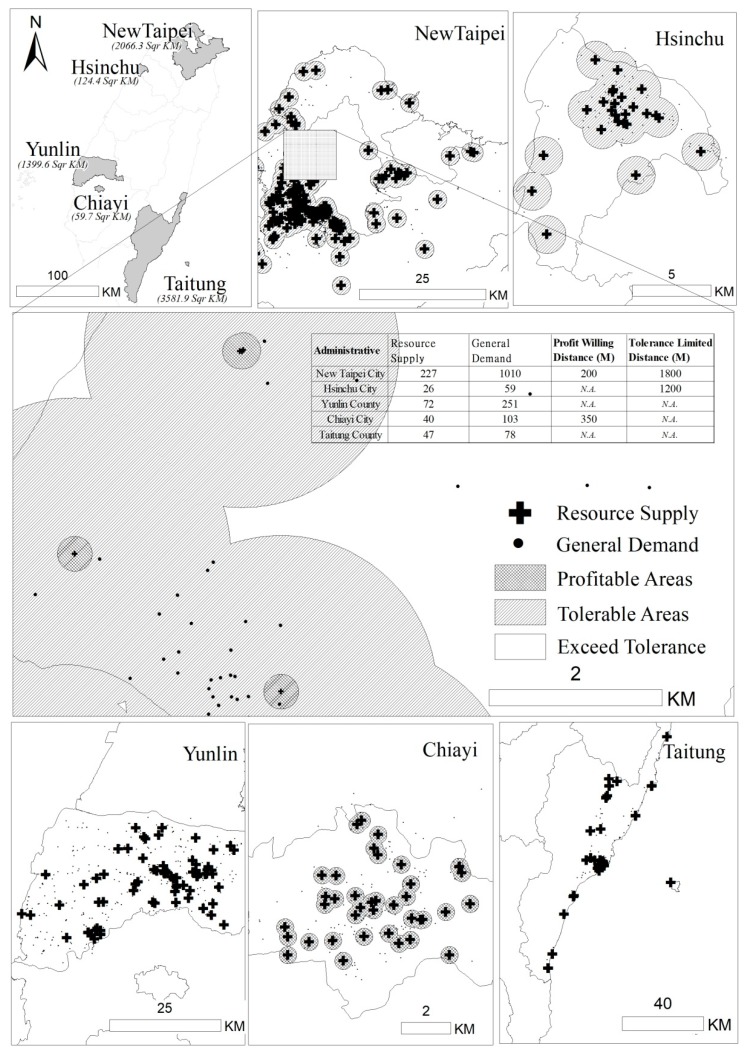
Schematic diagram of different situations of PWD and TLD: taking five cities and counties as examples.

**Table 1 ijerph-16-01754-t001:** Demographic characteristics of dementia cases in the current study.

Variables		Dementia Cases Living in the Community	Dementia Cases with General Demand ^a^	Dementia Cases with High Level of Need ^b^	*p*-Value ^ab^
		*n* = 9704	*n* = 6593	*n* = 3111	
		*n* (%)	*n* (%)	*n* (%)	
Age (mean ± SD)		78.22 ± 9.78	76.98 ± 9.93	82.10 ± 8.48	<0.001
	21–30	5 (0.1)	5 (0.1)	0	<0.001
	31–-40	30 (0.3)	26 (0.4)	4 (0.1)	
	41–50	117 (1.2)	102 (1.5)	15 (0.5)	
	51–60	403 (4.2)	349 (5.3)	54 (1.7)	
	61–70	1181 (12.2)	967 (14.7)	214 (6.9)	
	71–80	3432 (35.4)	2553 (38.7)	879 (28.3)	
	81–90	3975 (41.0)	2374 (36)	1601 (51.5)	
	91–100	553 (5.7)	215 (3.3)	338 (10.9)	
	101–110	8 (0.1)	2 (0.03)	6 (0.2)	
Gender	Male	3590 (37)	2456 (37.3)	1134 (36.5)	0.446
	Female	6114 (63)	4137 (62.7)	1977 (63.5)	
Severity of Disability	Mild	2729 (28.1)	2526 (38.3)	203 (6.5)	<0.001
	Moderate	3615 (37.3)	2686 (40.7)	929 (29.9)	
	Severe	734 (7.6)	432 (6.6)	302 (9.7)	
	Extremely serious	2626 (27.1)	949 (14.4)	2626(53.9)	

*p*-Value ^ab^: To compare the scores between the groups with general and high level of need; ^ab^: Definitions of general demand versus high level of need: ^a^ Dementia cases with general demand: the case’s domain scores are below the following cutoffs: Domain 1: 77.5, Domain 2: 78, Domain 3: 55, and Summary Score: 66.5. ^b^ Dementia cases with high level of need: The case’s domain scores are above these cutoffs [26].

**Table 2 ijerph-16-01754-t002:** Activity and participation functioning scores (WHODAS 2.0) of dementia cases in the current study.

Score	Dementia Cases Living in the Community	Median	Dementia Cases with General Demand ^a^	Median	Dementia Cases with High Level of Need ^b^	Median	*p*-Value ^ab^
	*n* = 9704		*n* = 6593		*n* = 3111		
	(mean ± SD)		(mean ± SD)		(mean ± SD)		
Summary score	65.74 ± 23.33	68.87	54.56 ± 19.60	56.52	89.43 ± 7.58	90.22	<0.001
D1 Cognition	72.13 ± 25.65	80.00	61.17 ± 23.89	60.00	95.35 ± 7.02	100	<0.001
D2 Mobility	60.61 ± 35.42	62.50	44.64 ± 31.62	43.75	94.45 ± 10.41	100	<0.001
D3 Self-care	57.70 ± 35.42	60.00	40.05 ± 29.16	40.00	95.11 ± 7.15	100	<0.001
D4 Getting along	70.47 ± 29.56	80.34	59.21 ± 28.86	58.33	94.35 ± 11.02	100	<0.001
D5 Life activities	83.06 ± 26.90	100.00	75.37 ± 29.45	90.00	99.37 ± 5.28	100	<0.001
D6 Participation	52.11 ± 25.74	50.0	43.33 ± 22.33	41.67	70.74 ± 22.36	75.00	<0.001

*p*-Value ^ab^: To compare the scores between the groups with general and high level of need; ^ab^: Definitions of general demand versus high level of need: ^a^ Dementia cases with general demand: the case’s domain scores are below the following cutoffs: D1: 77.5, D2: 78, D3: 55, and Summary score: 66.5. ^b^ Dementia cases with high level of need: The case’s domain scores are above these cutoffs [26].

**Table 3 ijerph-16-01754-t003:** Profiles of providers and dementia cases in 22 administrative areas in Taiwan.

City or County	Area	Providers ^a^	Dementia Cases Living in the Community ^b^	Dementia Cases with General Demand	Dementia Cases with High Level of Need	Ratio of Dementia Cases to Providers (b/a)
	km^2^	*n* (%)				
Taipei City	271.8	162 (9.1)	1564 (16.1)	996 (15.1)	568 (18.3)	9.65
New Taipei City	2052.6	227 (12.7)	1554 (16.0)	1010 (15.3)	544 (17.5)	6.85
Keelung City	132.8	38 (2.1)	105 (1.1)	81 (1.2)	24 (0.8)	2.76
Taoyuan City	1221.0	102 (5.7)	622 (6.4)	409 (6.2)	213 (6.9)	6.10
Hsinchu County	1427.5	48 (2.7)	153 (1.6)	84 (1.3)	69 (2.2)	3.19
Hsinchu City	104.2	26 (1.5)	90 (0.9)	59 (0.9)	31 (1.0)	3.46
Miaoli County	1820.3	41 (2.3)	194 (2.0)	122 (1.9)	72 (2.3)	4.73
Taichung City	2214.9	149 (8.4)	868 (8.9)	563 (8.5)	305 (9.8)	5.83
Changhua County	1074.4	90 (5.1)	416 (4.3)	292 (4.4)	124 (4.0)	4.62
Nantou County	4106.4	42 (2.4)	232 (2.4)	178 (2.7)	54 (1.7)	5.52
Yunlin County	1290.8	72 (4.0)	388 (4.0)	251 (3.8)	137 (4.4)	5.39
Chiayi County	1903.6	54 (3.0)	358 (3.7)	261 (4.0)	97 (3.1)	6.63
Chiayi City	60.0	40 (2.2)	152 (1.6)	103 (1.6)	49 (1.6)	3.80
Tainan City	2191.7	169 (9.5)	965 (9.9)	719 (10.9)	246 (7.9)	5.71
Kaohsiung City	2951.9	237 (13.3)	1005 (10.4)	737 (11.2)	268 (8.6)	4.24
Pingtung County	2775.6	98 (5.5)	394 (4.1)	279 (4.2)	115 (3.7)	4.02
Yilan County	2143.6	72 (4.0)	263 (2.7)	197 (3.0)	66 (2.1)	3.65
Hualien County	4628.6	40 (2.2)	181 (1.9)	126 (1.9)	55 (1.8)	4.53
Taitung County	3515.3	47 (2.6)	123 (1.3)	78 (1.2)	45 (1.5)	2.62
Penghu County	126.9	13 (0.7)	32 (0.3)	20 (0.3)	12 (0.4)	2.46
Kinmen County	151.7	9 (0.5)	42 (0.4)	27 (0.4)	15 (0.5)	4.67
Lienchiang County	28.8	6 (0.3)	3 (0.0)	1 (0.0)	2 (0.1)	0.50
Total	36,194.4	1782 (100)	9704 (100)	6593 (100)	3111 (100)	5.45

**Table 4 ijerph-16-01754-t004:** The profit willing distances (PWDs) and tolerance limited distances (TLDs) of 22 administrative areas in Taiwan.

City or County	Providers ^a^	Dementia Cases Living in the Community ^b^		Dementia Cases with General Demand		Dementia Cases with High Level of Need for	
	*n*	PWD (m)	TLD (m)	PWD (m)	TLD (m)	PWD (m)	TLD (m)
Taipei City	162	650	1200	350	1300	-	600
New Taipei City	227	200	2200	200	1800	-	950
Keelung City	38	300	1250	-	-	-	550
Taoyuan City	102	200	3250	250	1850	150	2250
Hsinchu County	48	200	1500	300	1200	-	-
Hsinchu City	26	150	-	-	1200	-	-
Miaoli County	41	150	2950	150	1000	-	-
Taichung City	149	150	3000	150	3000	150	2150
Changhua County	90	150	4750	-	-	-	-
Nantou County	42	150	3650	-	4050	-	-
Yunlin County	72	-	6250	-	-	-	-
Chiayi County	54	300	1200	400	700	-	-
Chiayi City	40	250	-	350	-	350	-
Tainan City	169	350	1200	300	1450	450	4500
Kaohsiung City	237	350	1600	300	1700	300	1600
Pingtung County	98	150	3600	150	2950	-	-
Yilan County	72	150	1200	-	1100	-	-
Hualien County	40	600	1700	550	1650	-	-
Taitung County	47	-	1300	-	-	-	900
Penghu County	13	-	-	-	-	-	-
Kinmen County	9	-	-	-	-	-	-
Lienchiang County	6	-	-	-	-	-	-

**Table 5 ijerph-16-01754-t005:** A summary of the PWDs and TLDs grouped by the needs of dementia cases.

The Status of PWD and TLD	All Dementia Cases		Dementia Cases with General Demand		Dementia Cases with High Level of Need	
	*n*	City or County	*n*	City or County	*n*	City or County
With both PWD and TLD	15	Taipei City	11	Taipei City	4	Taoyuan City
		New Taipei City		New Taipei City		Taichung City
		Keelung City		Taoyuan City		Tainan City
		Taoyuan City		Hsinchu County		Kaohsiung City
		Hsinchu County		Miaoli County		
		Miaoli County		Taichung City		
		Taichung City		Chiayi County		
		Changhua County		Tainan City		
		Nantou County		Kaohsiung City		
		Chiayi County		Pingtung County		
		Tainan City		Hualien County		
		Kaohsiung City				
		Pingtung County				
		Yilan County				
		Hualien County				
Only PWD	2	Hsinchu City	1	Chiayi City	1	Chiayi City
		Chiayi City				
Only TLD	2	Yunlin County	3	Hsinchu City	4	Taipei City
		Taitung County		Nantou County		New Taipei City
				Yilan County		Keelung City
						Taitung County
Neither PWD nor TLD	3	Penghu County	7	Keelung City	13	Hsinchu County
		Kinmen County		Changhua County		Hsinchu City
		Lienchiang County		Yunlin County		Miaoli County
				Taitung County		Changhua County
				Penghu County		Nantou County
				Kinmen County		Yunlin County
				Lienchiang County		Chiayi County
						Pingtung County
						Yilan County
						Hualien County
						Penghu County
						Kinmen County
						Lienchiang County

**Table 6 ijerph-16-01754-t006:** A comparison of the other measurements with PWD and TLD for policy planning of medical and long-term care.

Year	2005	2009	2009	2016	2015	Present Study
Author(s)	Wang and Luo	McGrail and Humphreys	Luo and Qi	Kilinc et al.	Lin et al.	Yen & Lin
Purpose	Assessing the accessibility of primary care in Illinois	Assessing the accessibility of primary care in rural areas in Victoria, Australia	Measuring the accessibility of primary care physicians	Assessing and measuring the accessibility and disparity in home care services	Developing new indices to compare nursing home care services in urban and rural areas	Using the PWD and TLD to examine the accessibility of dementia services and plan for resource allocation based on distributive justice
Method	2SFCA, factor analysis	2SFCA, closest facility analysis (a tool of network analysis)	E2SFCA, spatial cluster analysis	Revised 2SFCA, spatial cluster analysis	Spatial autocorrelation, regression	Spatial autocorrelation, join point analysis
Index used *	$R_{j}=\frac{S_{j}}{\sum_{i\in L_{j}}P_{i}}$ $A_{i}=\underset{j\in L_{i}}{\sum }R_{j}$	$R_{j}=\frac{S_{j}}{\sum_{i\in L_{j}}P_{i}}$ $A_{i}=\underset{j\in L_{i}}{\sum }R_{j}$	$R_{jk}=\frac{S_{jk}c_{k}}{\sum_{i\in Z_{j}}P_{i}d_{k}}$ $A_{ik}=\underset{j\in H_{i}}{\sum }R_{jk}$	Moran’s I	PWD TLD	Revised PWD and TLD
Index development & variables collected (level of complexity)	Spatial and non-spatial factors (e.g., individual’s age, gender, race, socioeconomic status and language skills) and characteristics of a population or area (e.g., land use, university graduation rate, ratio of single-parent families, state of unemployment, and ratio of occupational categories). Weighted factors are derived based on their eigenvalues and fuzzy logic.				Spatial (i.e., locations of service users and providers) and non-spatial factors (i.e., service availability, acuteness of service demand). The WHODAS 2.0 assessment results (as in the present study) or other measurements of health needs (target group).	

* *Rj* = Ratio of the service provider to the service user; *Sj* = Number of service providers; *Lj* = Location in a geographic area; *Pi* = Number of service users at location *i*; *Ai* = Accessibility of service providers to the service users at location *i*; *Li* = Locations of all service providers in a geographic area.

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
