# Peer review of "Justice in Dementia Care Resource Allocation: How Should We Plan for Dementia Services?"

_ijerph, 2019, doi:10.3390/ijerph16101754_

Round 1
Reviewer 1 Report
The paper describes the use of two new indicators (PWD and TLD) to assess service availability and access across the whole of Taiwan. The concepts of PWD and TLD are quite complex and I think that it would be more appropriate to describe them in the introduction rather than in section 2.3.2 where they are at present.
The paper seems in general to make sense and to produce evidence for why PWD and TLD might be good alternatives to other methods for describing service need and allocation. Presumably calculating these indices requires good national data to be available, so how applicable are these findings or these methods to other countries? If national data are poor, then I imagine that it will be difficult to make estimates of PWD or TLD.
Also I wondered whether distance is however a rather deceptive variable. In the maps, PWD and TLD are represented by neat circles whereas in reality catchment areas are likely to be uneven in shape. Also, presumably such things as the availability of transport may have an effect. In practice, it is the travelling time that seems more important than the actual distance. Perhaps this requires some more discussion.
I note that dementia cases are divided into two categories, those with 'general demand' and those with 'acute demand'. I would take issue with the use of the word 'acute' here, as this usually means that there is medical urgency, and I think the people in this category are characterised more by a high level of need rather than being in a medically urgent state. I think a different term should be used, otherwise it is misleading for the reader.
There are one or two typos or errors, e.g. Table 1, column 2: should be 'mild' not 'mile'; and in line 190 it should be 'mean' not 'meaning'.
Author Response
Revision Note
International Journal of Environmental Research and Public Health
Manuscript ID: ijerph-499884
Type of Manuscript: Article
Title: Justice in Dementia Care Resource Allocation: How Should We Plan for Dementia Services?
Dear IJERPH Editorial Office
We would like to thank the review committees for their valuable suggestions. We have revised the manuscript based on these suggestions and related to the full-text content in red color (include the paragraphs of Author Contributions and Funding). Our responses and the pagination for the revisions in the revised manuscript are listed as follows with a point-by-point response and appendix to the reviewers' comments:
Reviewer #1: | Response/revised |
The paper describes the use of two new indicators (PWD and TLD) to assess service availability and access across the whole of Taiwan. The concepts of PWD and TLD are quite complex and I think that it would be more appropriate to describe them in the introduction rather than in section 2.3.2 where they are at present. | Thank you for your suggestion, we have added the description on page 2 line 59-65. |
The paper seems in general to make sense and to produce evidence for why PWD and TLD might be good alternatives to other methods for describing service need and allocation. Presumably calculating these indices requires good national data to be available, so how applicable are these findings or these methods to other countries? If national data are poor, then I imagine that it will be difficult to make estimates of PWD or TLD. | Thank you for the reviewer’s asking about the application in other countries limitations of PWD and TLD. The concepts of PWD and TLD are very important particularly in some countries, their public medical services fee paid by the government don’t include the cost due to the distance which must be paid by providers or user-self. The distance cost indeed influences the willingness of providers and the accessible of national welfare services. Indeed, these two indicators do have some limitations in application. From the findings of this study, if the number of service users (patients) is too small, it is difficult to find PWD and TLD by joinpoint analysis. We have added the description on p. 14 lines 296-302 and line 310-313. |
Also, I wondered whether the distance is however a rather deceptive variable. In the maps, PWD and TLD are represented by neat circles whereas in reality catchment areas are likely to be uneven in shape. Also, presumably such things as the availability of transport may have an effect. In practice, it is the travelling time that seems more important than the actual distance. Perhaps this requires some more discussion. | Thanks for your consideration carefully, we agreed that the traveling time is more appropriate than the nearest distance to be used in this study. We will add the discussion in the contents on p. 14 line 300-302.
|
I note that dementia cases are divided into two categories, those with 'general demand' and those with 'acute demand'. I would take issue with the use of the word 'acute' here, as this usually means that there is medical urgency, and I think the people in this category are characterised more by a high level of need rather than being in a medically urgent state. I think a different term should be used, otherwise it is misleading for the reader. There are one or two typos or errors, e.g. Table 1, column 2: should be 'mild' not 'mile'; and in line 190 it should be 'mean' not 'meaning'. | Thanks for your consideration carefully, we have replaced “acute demand” with “high level of need” in the manuscript (all the words in red). And, all the errors have been revised in red. Thank you very much. |

Reviewer 2 Report
The paper is outlines the topic area lucidly and cogently. The objective of the study is distinctive and the process of its exploration well articulated. The quality of the writing is very high throughout and this provides a platform for some convincing evaluations. The tables are also well configured and presented. My comments are limited to some minor suggestions:
Could the conclusion offer a more definite synthesis of what was achieved in this study? Currently the conclusion is a fairly general summary of the process. Even though the concluding section is brief, it could still offer a more positive summary.
The initialism PWD is used in the paper in relation to 'profit willing distance' - and this is entirely appropriate. However, this initialism is sometimes used in dementia research to refer to a Person With Dementia. This could jar a little with readers who are familiar with this latter use of the initialism. A brief acknowledgement of this alternative usage (as a parenthetical point in the introductory section perhaps) should overcome this minor issue.
Author Response
Revision Note
International Journal of Environmental Research and Public Health
Manuscript ID: ijerph-499884
Type of Manuscript: Article
Title: Justice in Dementia Care Resource Allocation: How Should We Plan for Dementia Services?
Dear IJERPH Editorial Office
We would like to thank the review committees for their valuable suggestions. We have revised the manuscript based on these suggestions and related to the full-text content in red color (include the paragraphs of Author Contributions and Funding). Our responses and the pagination for the revisions in the revised manuscript are listed as follows with a point-by-point response and appendix to the reviewers' comments:
Reviewer #2: | |
Could the conclusion offer a more definite synthesis of what was achieved in this study? Currently the conclusion is a fairly general summary of the process. Even though the concluding section is brief, it could still offer a more positive summary. | Thanks for reviewer’s suggestions. We have substantially revised the conclusion paragraph in the manuscript. p. 14 line 307-315.
|
The initialism PWD is used in the paper in relation to 'profit willing distance' - and this is entirely appropriate. However, this initialism is sometimes used in dementia research to refer to a Person With Dementia. This could jar a little with readers who are familiar with this latter use of the initialism. A brief acknowledgement of this alternative usage (as a parenthetical point in the introductory section perhaps) should overcome this minor issue. | Thanks for the reviewer’s suggestions. We have added the brief instruction in line 62-63.
|
